# Genomic and Transcriptomic Characterisation of Response to Neoadjuvant Chemoradiotherapy in Locally Advanced Rectal Cancer

**DOI:** 10.3390/cancers12071808

**Published:** 2020-07-06

**Authors:** Sinead Toomey, Jillian Gunther, Aoife Carr, David C. Weksberg, Valentina Thomas, Manuela Salvucci, Orna Bacon, El-Masry Sherif, Joanna Fay, Elaine W. Kay, Katherine M. Sheehan, Deborah A. McNamara, Keith L Sanders, Geena Mathew, Oscar S. Breathnach, Liam Grogan, Patrick G. Morris, Wai C. Foo, Yi-Qian N. You, Jochen H. Prehn, Brian O’Neill, Sunil Krishnan, Bryan T. Hennessy, Simon J. Furney

**Affiliations:** 1Medical Oncology Group, Department of Molecular Medicine, Royal College of Surgeons in Ireland, Dublin, Dublin 9, Ireland; sineadtoomey@rcsi.ie (S.T.); aoifecarr@rcsi.ie (A.C.); 2Department of Radiation Oncology, The University of Texas MD Anderson Cancer Center, Houston, TX 77030, USA; JGunther@mdanderson.org (J.G.); davidweks@gmail.com (D.C.W.); KLSanders@mdanderson.org (K.L.S.); geenag@mdanderson.org (G.M.); 3UPMC Pinnacle, Harrisburg, PA 17101, USA; 4Genomic Oncology Research Group, Department of Physiology and Medical Physics, Royal College of Surgeons in Ireland, Dublin, Dublin 2, Ireland; valentinathomas@rcsi.ie; 5Centre for Systems Medicine, Department of Physiology and Medical Physics, Royal College of Surgeons in Ireland, Dublin, Dublin 2, Ireland; manuelasalvucci@rcsi.ie (M.S.); ornabacon@rcsi.ie (O.B.); JPrehn@rcsi.ie (J.H.P.); 6Department of Surgery, Our Lady of Lourdes Hospital Drogheda, Co. Louth, Ireland; selmasry@rcsi.ie; 7Department of Surgery, Beaumont Hospital, Dublin, Dublin 9, Ireland; deborahmcnamara@rcsi.ie; 8Department of Pathology, Royal College of Surgeons in Ireland, Dublin, Dublin 9, Ireland; joannafay@rcsi.ie (J.F.); elainewkay@gmail.com (E.W.K.); ksheehan@rcsi.ie (K.M.S.); 9Department of Medical Oncology, Beaumont Hospital, Dublin, Dublin 9, Ireland; osbreathnach@beaumont.ie (O.S.B.); liamgrogan@beaumont.ie (L.G.); patrickmorris@beaumont.ie (P.G.M.); 10Department of Pathology, The University of Texas MD Anderson Cancer Center, Houston, TX 77030, USA; WFoo@mdanderson.org; 11Department of Surgical Oncology, The University of Texas MD Anderson Cancer Center, Houston, TX 77030, USA; YNYou@mdanderson.org; 12Department of Radiation Oncology, St. Luke’s Radiation Oncology Centre, Beaumont Hospital, Dublin 9, Ireland

**Keywords:** locally advanced rectal cancer, neoadjuvant chemoradiotherapy, whole exome sequencing, RNA-seq, microbiome, microsatellite instability

## Abstract

Standard treatment for locally advanced rectal cancer (LARC) is neoadjuvant chemoradiotherapy (NACRT), followed by surgical resection. However, >70% of patients do not achieve a complete pathological response and have higher rates of relapse and death. There are no validated pre- or on-treatment factors that predict response to NACRT besides tumour stage and size. We characterised the response of 33 LARC patients to NACRT, collected tumour samples from patients prior to, during and after NACRT, and performed whole exome, transcriptome and high-depth targeted sequencing. The pre-treatment LARC genome was not predictive of response to NACRT. However, in line with the increasing recognition of microbial influence in cancer, RNA analysis of pre-treatment tumours suggested a greater abundance of *Fusobacteria* in intermediate and poor responders. In addition, we investigated tumour heterogeneity and evolution in response to NACRT. While matched pre-treatment, on-treatment and post-treatment tumours revealed minimal genome evolution overall, we identified cases in which microsatellite instability developed or was selected for during NACRT. Recent research has suggested a role for adaptive mutability to targeted therapy in colorectal cancer cells. We provide preliminary evidence of selection for mismatch repair deficiency in response to NACRT. Furthermore, pre-NACRT genomic landscapes do not predict treatment response but pre-NACRT microbiome characteristics may be informative.

## 1. Introduction

Globally, colorectal cancer (CRC) is the third most common type of cancer, making up about 10% of all cases, and the fourth leading cause of cancer death. CRC is more common in developed countries where more than 65% of cases and approximately 60% of all deaths occur [1]. One third of all colorectal cancers are located in the rectum. Histologically, over 95% of rectal cancers are adenocarcinomas and the remaining 5% consist of squamous cell carcinomas, lymphomas and sarcomas [2]. The majority of rectal cancer patients present with locally advanced rectal cancer (LARC), defined by stage T3 or T4 and/or a lymph node positive tumour. The standard treatment for LARC is neoadjuvant chemoradiotherapy (NACRT), whereby the patient undergoes radiation therapy and 5-Fluorouracil (5-FU) or capecitabine for 5–6 weeks, followed by surgical resection of the tumour [3]. Multiple studies have shown direct correlation between NACRT and reduced local relapse rates and improved patient survival rates compared to adjuvant treatment [4,5]. The benefits of NACRT are most marked when a pathologic complete response (pCR) is achieved (i.e., when no remaining tumour is identified upon pathological examination of the subsequent surgical specimen) [6]. Currently only 15–27% of LARC patients achieve pCR, and the remainder have residual disease, varying from a few scattered tumour cells to large islands of resistant tumour. As a result, these patients have subsequent higher relapse and death rates [6].

We and others have shown that activating mutations in a limited number of genes in the phosphatidylinositol 3-kinase (PI3K) and MAP kinase (MAPK) signalling pathways can modulate treatment responsiveness and clinical outcomes in LARC [7,8]. Large-scale sequencing studies have revealed the complex genomic landscape of CRC, [9,10] however information on the genomic landscape of LARC is limited and little is known regarding the clonal evolution of LARC and the impact of NACRT on the genomic landscape of residual tumour at surgery.

To determine the association between the genomic evolution of LARC and clinical outcomes, we have performed whole exome and high-depth targeted sequencing on tumour samples collected from patients prior to, during and after NACRT. 

## 2. Results

### 2.1. Patient Characteristics

Thirty-three locally advanced rectal cancer patients were included in the study. Seventeen (51.5%) patients were male and 16 (48.5%) patients were female. The median age at diagnosis was 54 years (range 42–79). Regarding tumour location, 14 patients (39.4%) had a tumour located 5 cm from the anal verge (low rectal cancer), 14 patients (42.4%) had tumours that were within 6–10 cm of the anal verge (middle rectal cancer), and 6 patients (18.2%) had tumours that were within 11–15 cm of the anal verge (high rectal cancer). All patients received NACRT consisting of 5-FU (5-Flurouracil) or capecitabine and 50.4 Gy radiotherapy in 28 fractions. Once patients completed their course of treatment, they underwent surgical resection. Anterior resection (AR) was performed in 20 cases (60.6%), abdominoperineal resection (APR) in 5 cases (15.1%), coloanal anastomosis (CAA) in 5 cases (15.1%), and transanal excision (TAE) in 1 case (3%). Two patients (6.1%) did not undergo surgery. In the patients who underwent surgical resection following NACRT, 5 patients (15.1%) had complete tumour regression (RCPath A), 12 patients (36.4%) had occasional microscopic foci of residual tumour (RCPath B), and 14 patients (42.4%) had little or no obvious tumour regression (RCPath C).

### 2.2. Sequencing Strategy Summary

Beaumont Hospital tumour and normal samples underwent whole exome sequencing (Figure 1A; Appendix A; mean on-target coverage 83X). Tumour samples underwent higher depth panel DNA sequencing (mean on-target coverage 368X) and RNA-seq (60M reads). MDACC tumour and normal samples underwent whole exome sequencing (mean on-target coverage 34X). Tumour samples underwent higher depth panel DNA sequencing (mean on-target coverage >5000X).

### 2.3. The Pre-NACRT LARC Genome Landscape

Whole exome sequencing of the pre-NACRT LARC genomes from 28 patients revealed that the vast majority of the tumours contained likely functional mutations (SNVs or indels) in known colorectal cancer genes (27/28 genomes; Figure 1B). *APC* was mutated in 25 samples (89%), *TP53* in samples 20 samples (71%), and *KRAS* in 9 samples (32%). Comparison with samples annotated as rectal (C20) in the TCGA Rectum Adenocarcinoma (READ) cohort (n = 72) shows broadly similar frequencies of mutations in known colorectal cancer genes in our smaller cohort. Notably our cohort shows an absence of *NRAS* driver mutations (TCGA C20: 6.6%), and a higher frequency of *BRAF* mutations (LARC: 10.7%; TCGA C20: 1.6%). Tumour mutation burden and mutational signature analysis were estimated from the exome sequencing data and identified two likely MSI samples (Patients 2 and 15; Figure 1B), consistent with the known lower frequency of MSI tumours in the rectum compared to the colon. Mutational signature analysis indicated that many of the mutations are estimated to be due to an age-related signature (COSMIC signature 1; Figure 1B). Somatic copy number analysis revealed alterations characteristic of colorectal cancer, including gains at chromosome 7, 8q, 13q, 20p and 20q, and losses at chromosome 4, 8p, 17p, 18p and 18q (Figure 1C,D; Appendix A; Appendix A) [11,12]. In terms of specific genes, we saw recurrent gains of genes like *AUKRA* and *SNA1* and recurrent losses of genes such as *TP53* and *SMAD4*.

### 2.4. Baseline Genomic Predictors of Response to NACRT

Of the patients who underwent pre-NACRT tumour WES and surgical resection following NACRT, five patients had complete tumour regression (RCPath A; Figure 1 and Figure 2; Good), 11 patients had occasional microscopic foci of residual tumour (RCPath B; Intermediate), and 10 patients had little or no obvious tumour regression (RCPath C; Poor). Two patients did not have surgery, so no RCPath grade was available. Analysis of summary genomic burdens showed no statistically significant differences between the response groups in SNV number, indel number or overall copy number burden (Kruskal–Wallis rank sum test; Figure 2A–C). Enrichment analysis of mutational frequency with response groups did not reveal any significantly different mutated genes between the groups and the somatic copy number alterations were largely similar (Figure 2D). We compared regions of copy number alteration by response group (Good versus Poor, and Good versus Intermediate and Poor) [13], which highlighted one region on chromosome 6 and one region on chromosome 7, respectively (Appendix A). We conducted transcriptome sequencing of 16 pre-treatment tumours enabling microbiome analysis, which highlighted a trend in lower *Fusobacteria* abundance in the good response group compared to the intermediate/poor responders (Kruskal-Wallis *p*-value = 0.09; Figure 2E; Appendix A). Estimation of differences in proportion of different immune cell types between the response groups did not identify a significant association with response group (Figure 2F; Kruskal–Wallis test NS). A recent study described mutations in genes related to homologous recombination (HR) in LARC patients evaluated according to response to therapy [14]. We identified somatic mutations HR-related genes in our patient cohort (Appendix A), and calculated the contribution of COSMIC mutational signature 3 (HR-associated) to each tumour genome (Appendix A). However, there is no significant difference between the groups for signature 3 SNVs (Kruskal-Wallis test *p*-value = 0.3557) in our cohort of patients.

### 2.5. Early Genomic Landscape Drift (Pre-NACRT vs. on-NACRT)

In total, 12 patients had both pre-NACRT and on-NACRT tumours sequenced (Appendix A). A further four patients had both pre-NACRT and post-NACRT resection samples sequenced. To investigate tumour evolution in response to therapy we tracked driver mutations (*APC*, *TP53*, *KRAS*, *PIK3CA*, *BRAF*) in tumours with pre- and on-NACRT samples (Appendix A). We did not identify the emergence of any new driver mutations in these genes in the on-NACRT samples (even with the high-depth panel sequencing data). For several patients we obtained two pre-NACRT and/or on-NACRT biopsies enabling us to survey intra-tumour heterogeneity. In matched pre-NACRT biopsies (Patients 6, 7, 8, 10, 20, 23, 24) all of the driver mutations are present in both samples, albeit at differing variant allele frequencies (VAFs; Appendix A). Similar findings were noted in matched on-NACRT samples (Patients 6, 8, 9, 10, 21, 24). In addition, there is no difference in tumour mutational burden between the pre-NACRT vs. on-NACRT samples (Wilcoxon rank sum test NS).

### 2.6. Late Genomic Landscape Drift (Pre-NACRT vs. Post-NACRT)

For 12 patients (five Intermediate and seven Poor responders—Appendix A), we had pre-NACRT and post-NACRT (surgical resection) samples. Although some of the post-NACRT samples had low tumour content (e.g., patients 7, 8 and 22) we identified the vast majority of the driver mutations present pre-NACRT in the post-NACRT tumours. In patient 24 we do not see evidence of the *TP53* mutation present in the pre- and on-NACRT tumours, and the relative VAF of the *PIK3CA* mutation is much lower. We did not detect the emergence of any known colorectal cancer driver mutations in the post-NACRT tumours. A novel *HRA*S p. G13D mutation was detected in the post-NACRT tumour of patient 23; however, this mutation has not previously been described in rectal cancer. There is no difference in tumour mutational burden between the pre-NACRT vs. post-NACRT samples (Wilcoxon rank sum test NS). These results point to a general inertia in tumour evolution at the genomic level through the course of NACRT.

For four patients, we were able to conduct transcriptome sequencing of matched pre-NACRT and post-NACRT tumours. Interestingly, when we classified the tumours by consensus molecular subtype [15] (CMS), there was discordance between pre-NACRT and post-NACRT CMS for three patients (patients 14, 15 and 16—all intermediate responders) and concordance for the remaining patient (patient 28—poor responder; Appendix A). Immune cell type analysis revealed a slight increase in M1 macrophages in the post-NACRT tumours (Wilcoxon rank sum test *p* value = 0.02857; Appendix A).

### 2.7. Selection for Microsatellite Instability

In two cases (Patients 15 and 25), we see selection/emergence of a mutation in the *MSH2* gene after exposure to NACRT. In patient 15 (Intermediate response; Figure 3A), a *MSH2* 1bp deletion causes a frameshift mutation (p. P618*), which is present at a VAF of 0.05 in the pre-NACRT sample, rising to 0.22 in the post-NACRT sample. The VAFs of the other driver mutations *APC* and *BRA*F are almost identical in the two samples; however, the *PIK3CA* mutation is present at different VAFs (PRE: 0.08; POST: 0.21). The higher *MSH2* VAF is also reflected in an increased mutational burden in the post-NACRT sample: SNV (Figure 3A) and indel burden (PRE: 146, POST: 361). Of note, this patient is one of the patients for whom we had pre-NACRT and post-NACRT RNA-seq data (see previous section). Consensus molecular subtyping showed a change in subtypes from the biopsy (CMS3) to the resection (CMS1) sample. CMS1 has been associated with defective DNA mismatch repair [15]. In patient 25 (Poor response; Figure 3B) a *MSH2* 2bp deletion causes a frameshift mutation (p. R219*) in the on-NACRT sample. This mutation was not detected in the pre-NACRT sample and this is evident in the relative mutational burdens: SNV burden (Figure 3B) and indel burden (PRE: 21, ON: 1713, POST: 305).

### 2.8. Development of an Independent Primary Tumour

In one case (Patient 12), in addition to the biopsy sample, a further tumour was resected from this patient at a later date. We conducted panel sequencing of this tumour identifying several driver mutations not present in the original tumour, including *TP53* V122*, *PIK3CA* E542K, and *BRAF* K601E. In addition, the *TP53* and *APC* mutations in the original tumour were absent indicating that this tumour had likely arisen independently.

## 3. Discussion

Genomic alterations and response to NACRT in LARC are poorly understood. Our WES analysis revealed mutation frequencies that were in line with those determined in the TCGA rectal cohort [9]. Activation of the Wnt signalling pathway, predominantly through mutations in the *APC* gene, is a key oncogenic driver in most colorectal cancers [9,16,17]. In vitro activation of the Wnt signalling pathway mediates chemoradiotherapy resistance in colorectal cancer cell lines, [18] however our dataset did not demonstrate any dynamic alteration in frequency of *APC* mutations over the course of NACRT among all patients or as a function of TRG response. *TP53* mutations occur later in colorectal cancer development and are particularly common in non-hypermutated tumours [9]. Some studies have shown an association between mutations in *TP53* and resistance to radiotherapy [19,20] however our data do not support this in the context of NACRT. *KRAS* mutations occur early in CRC development [21] and the role of *KRAS* mutations in response to NACRT in rectal cancer is conflicting [8,22,23,24]. We did not identify any difference in response to NACRT between patients with *KRAS* wild-type and mutant tumours, nor did our data validate the enrichment of concurrent *KRAS*/*TP53* mutations in non-responders reported recently [23].

Spatial and temporal intra-tumour genomic heterogeneity (ITGH) is a common feature of many cancers [25]. Studies investigating ITGH in non-metastatic rectal cancers are limited with varying results [26,27,28]. We found a high concordance of driver mutations between pre-, on- and post-NACRT treatment samples. In all but one patient the vast majority of driver mutations present in the pre-NACRT tumours were also present in the post-NACRT tumours, despite reduced tumour cellularity in many post-NACRT tumours. Furthermore, other than *HRAS* G13D, we did not identify any new colorectal cancer driver mutations in the tumours during or post-NACRT. We also determined the degree of spatial ITGH in patients where we obtained two pre-NACRT and/or on-NACRT biopsies. Although any driver mutations identified were present in both samples, variant allele frequencies differed between the two samples, a potentially interesting finding as spatial ITGH may be associated with poor prognosis in several cancer types [29].

Consistent with other reports in rectal cancer but surprisingly different from other cancers, NACRT does not appear to exert a selection pressure on heterogeneous genomically unstable cancer cell populations to enrich for chemo-/radio-resistant clones, and notably not even in non-responders. Genomic evolution at the mutational level is not apparent; however, in two patients, microsatellite instability developed or was selected for during/after exposure to NACRT. Rectal cancers are known to have a lower frequency of MSI tumours than colon tumours [30], so the potential development of MSI in some rectal cancer patients during NACRT may be clinically relevant, given that patients with MSI-high tumours are likely to be good candidates for immunotherapy [31]. Recent research has suggested a role for adaptive mutability to targeted therapy in colorectal cancer cells [32]. Here we provide preliminary evidence of selection for increased mutation rate in response to NACRT.

Copy number analysis of pre-NACRT tumours revealed alteration characteristic of colorectal cancer. We found that chromosome 20 underwent gains most frequently and chromosome 18 copy number losses. The copy number of *AURKA* (Aurora Kinase A) on chromosome 20 is increased in 20/28 samples in our cohort. AURKA plays a role in mitosis and reduces the transcriptional activity of P53 via P53 stability [33]. Overexpression and amplification of Aurora kinases in cancers has been associated with poor prognosis [34]. Inhibition of AURKA has been shown to increase chemosensitivity and radiosensitivity in vitro [35], and in vivo [36]. Thus, AURKA may be a novel target in LARC that could help to improve NACRT response rates. Aurora kinase inhibitors are being investigated in clinical trials at present. Also present on chromosome 20 is *WISP2* (WNT1-inducible-signaling pathway protein 2) with high copy number in many samples in our cohort. As proteins downstream of WNT/beta-catenin, WISPs are implicated in colorectal cancer invasion and motility. Increased copy numbers of *EEF1A2* (eukaryotic translation elongation factor 1 alpha 2), also on chromosome 20, were also observed in many of our samples pre-, on- and post-NACRT. EEF1A2 is a protein translation factor that has been implicated in tumorigenesis in several cancers where it enhances cell growth and inhibits apoptosis. Finally, consistent with the dependence of colorectal carcinogenesis on WNT pathway activation in many instances, high copy numbers of *SNAI1* (Snail1), mapped to chromosome 20 as well, are seen in many samples in our cohort. Snail1 plays a pivotal role in epithelial-to-mesenchymal transition and metastasis. These trends seem consistent with the finding that copy numbers of chromosome 20 were lowest in good responders, intermediate in intermediate responders, and highest in poor responders (Figure 2D).

Overall, our results do not show any association of any particular gene with response or any statistically significant differences in good, intermediate and poor responders to NACRT in SNV number, indel number or overall copy number burden. Therefore, it is possible that predictors of response to NACRT may not exist in the genome of cancer cells, but may occur elsewhere. Recently, Kamran et al. demonstrated that while NACRT was not associated with increased mutational burden or a change in driver mutation frequency, non-response to NACRT was associated with reduced CD4/CD8 T-cell infiltration and a post-NACRT M2 macrophage phenotype [23]. Although we did not find differences between M2 macrophages between pre- and post-NACRT tumours, we identified an increase in M1 macrophages in the post-NACRT tumours (*p* = 0.0287). M1 macrophages which have been linked to a pro-inflammatory response are associated with good disease course [37].

In patients where we were able to conduct transcriptome sequencing of matched pre- and post-NACRT tumours, the consensus molecular subtype (CMS) changed after NACRT in 3/4 patients, all of whom were intermediate responders. 5-FU-based chemotherapy has been associated with emergence of a mesenchymal tumour phenotype [38] as occurred in our study, and previous studies have demonstrated the treatment-resistant nature of mesenchymal colorectal tumours [39,40].

Finally, we noted a trend towards decreased *Fusobacteria* abundance in baseline tumour biopsies in the good response group compared to the intermediate/poor responders. *Fusobacterium nucleatum* has been shown to be causative of colorectal cancer and is associated with more aggressive tumours and a higher mortality [41,42]. To our knowledge, our study is the first to suggest that tumour *Fusobacteria* may be associated with early treatment resistance as shown by TRG following NACRT in LARC, albeit in a small number of patients.

Relatively small patient numbers limit our study’s power to detect small genomic landscape changes and larger studies are needed to support our findings. Furthermore, the study was conducted on two patient cohorts deriving from two different centres, with different targeted sequencing strategies. Interestingly, we found very few additional mutations by panel sequencing compared to exome sequencing, therefore the impact of this on our results is likely to have been negligible. In addition, allele frequencies have not been normalised to tumour cellularity, so any decrease in allele frequency observed over the course of NACRT could potentially be related to a decrease in tumour cellularity.

Despite these limitations, useful insights can be obtained from our study, which is the largest of its kind to date in LARC, including the potential clinical utility of testing LARC patients for genomic and transcriptomic aberrations.

## 4. Materials and Methods

### 4.1. Rectal Cancer Tumour Samples

Informed consent was obtained from patients and ethical approval was obtained for the study (reference: 08/62) at Beaumont Hospital and the M.D. Anderson Cancer Centre. Fresh frozen locally advanced rectal adenocarcinoma pre- and on-treatment biopsy samples, post-treatment surgical samples, and matched normal samples from 33 patients were obtained from tumour banks under the auspices of Institutional Review Board-approved protocols at Beaumont Hospital (BH, Dublin, Ireland) and M.D. Anderson Cancer Centre (MDACC, Houston, Texas, USA). Normal samples were obtained from a distant site in the rectal tract that was macroscopically unaffected and disease free (BH, Doblin, Ireland) or from blood (MDACC, Houston, Texas, USA). LARC was defined by T3/4 and/or node-positive staging for tumours within 15 cm of the anal verge on rigid sigmoidoscopy without evidence of distant metastasis at diagnosis. The study group comprised 33 patients who had been diagnosed with LARC and who were treated with NACRT, between 2010 and 2016. Surgery was performed 8 weeks after the end of NACRT. Pathologic stage was determined using the either the American Joint Committee on Cancer, 7th edition or the Union for International Cancer Control TNM classification. pCR was defined as the absence of any residual invasive tumour in the surgical resection specimen, including the regional lymph nodes. Treatment response to chemoradiotherapy was classified as good, intermediate or poor based on the Royal College of Pathologists tumour regression grading (TRG) system, which was used clinically at the time that the samples were collected. Tumours were classified as having a good response if they achieved complete tumour regression (RCPath A); intermediate response if there was partial tumour regression (RCPath B); and poor response if tumours were upstaged or had no tumour regression (RCPath C).

### 4.2. Overview of Patient Cohort and Sequencing Strategy

#### 4.2.1. Sample Processing

All tumour samples and normal samples were snap frozen. Fresh-frozen samples were divided into three segments and a piece of tissue from the top and bottom part of each segment was sectioned, stained with haematoxylin and eosin and extensive pathological review was carried out by a trained pathologist to ensure there was a minimum of 30% tumour within the segment of the tumour samples and no tumour present in the normal samples. DNA was extracted from the tumour and normal samples using an AllPrep DNA mini kit (Qiagen, Hilden, Germany), and from whole blood samples using a DNA blood mini kit (Qiagen), according to the manufacturers protocol. DNA was quantified by Qubit fluorometer (Invitrogen, Carlsbad, CA, USA) and DNA integrity was examined by agarose gel electrophoresis.

#### 4.2.2. Whole Exome Sequencing

BH: For each tumour and matched normal tissue or blood sample, exome capture was performed on 2µg DNA using the Agilent SureSelect Human All Exome V3 kit, according to the manufacturers protocol (Agilent Technologies, Santa Clara, CA, USA), or SeqCap_EZ_Exome_v3. Samples were sequenced to a mean coverage of 83x using 91bp paired end reads on the Illumina HiSeq 2000.

#### 4.2.3. MDACC

Illumina-compatible indexed libraries were prepared from 500 ng of Biorupter Ultrasonicator (Diagenode)-sheared gDNA using the KAPA Hyper Library Preparation Kit (Kapa Biosystems, Wilmington, MA, USA). Library quality was assessed using the TapeStation High Sensitivity DNA Kit (Agilent Technologies). Exome capture of the library pool was performed using the NimbleGen SeqCap EZ Exome kit V3.0 (Roche-Nimblegen, Madison, WI, USA). Sequencing was then performed in one lane of the HiSeq3000 Sequencer (Illumina Inc., San Diego, USA) using the 76 nt paired end format.

#### 4.2.4. Whole Exome Sequencing Data Analysis

The quality of FASTQ files generated was determined using FastQC, and adapter and primer sequences and low quality 3′ end reads were trimmed off using Trimmomatic [43]. Remaining reads (with a minimum length of 90 bases) were aligned to the hg19 reference genome using BWA [44]. PCR duplicates were marked using Picard Tools (http://broadinstitute.github.io/picard) and InDel realignment and base quality recalibration were conducted with GATK v3 [45]. Somatic single nucleotide variants (SNVs) were identified with mutation calling algorithms MuTect v1 [46] and somatic indels with VarScan 2 [47]. Identified somatic variants were annotated using Variant Effect Predictor [48] and variants within the targeted capture genes were kept for further analysis.

#### 4.2.5. Copy Number Analysis

Copy number alterations and tumour purity and ploidy were estimated using FACETS [49] and EXCAVATOR2 [50]. Significant regions of copy number alteration were identified with GISTIC2 using default parameters at a q value threshold of 0.25. [12]. Copy number alteration analysis between response groups was conducted using CoNVaQ at a q value threshold of 0.25 [13].

#### 4.2.6. Deep Sequencing of Cancer Gene Panels

A panel of 151 cancer genes (BH; ClearSeq Comprehensive Cancer Panel, Agilent Technologies) or 48 cancer genes (MDACC; Amplicon Cancer Panel, Illumina) was captured from samples with sufficient remaining tumour DNA and sequenced, following the manufacturers protocol.

#### 4.2.7. Mutational Signature Analysis

Mutational signature analysis was performed to inform on the exposures and biological history of a cancer. Mutational signatures were identified from SNVs using the R package deconstructSigs v1.8 [51] based on the pan-cancer catalogue of 30 signatures referenced in the COSMIC database (https://cancer.sanger.ac.uk/cosmic/signatures).

#### 4.2.8. Transcriptome Sequencing Data Analysis

##### Gut Microbiome Analysis

Tumour tissue-associated gut microbiota were identified from RNA data using PathSeq v2.0 [52], available from the Genome Analysis Toolkit (GATK) v4 (http://www.broadinstitute.org/software/pathseq/).

RNA-Seq data: quantification of gene expression, normalisation and gene ID conversion. Gene counts of tumour samples were generated from RNA-seq data using Kallisto [53]. Normalisation of RNA counts was performed with DESeq2 v1.24 [54], using rlog and the “~patient_id” formula. Ensembl gene IDs were mapped to Entrez and Symbol IDs using biomaRt v2.40 [55].

##### Molecular Subtyping and Tumour Immune Infiltration

Molecular subtyping of tumours based on gene expression profiles was performed using two R implemented classification systems: the CMS classifier v1 [15], and the CRIS classifier v1 [56]. Tumour immune contexture was analysed for each sample applying the quanTIseq computational pipeline [57], which uses RNA-Seq data to quantify the fractions of ten immune cell types (B cells, classically activated macrophages M1, alternatively activated macrophages M2, neutrophils, natural killer cells, CD4^+^ T cells, CD8^+^ T cells, regulatory T cells, monocytes and dendritic cells) in heterogeneous tissues.

## 5. Conclusions

Importantly, our study gives an unprecedented view of the LARC tumour genome by the investigation of pre-treatment, during NACRT and post-resection tumour samples. Our data suggest that predictors of response to NACRT are not likely to be determined simply by genomic analysis; overall response could be determined by a multitude of factors that are yet to be elucidated.

However, we present several promising avenues in the quest to understand response to NACRT in LARC so as to enable the eventual selection of patients to avoid the morbidity of radical surgical resection. RNA analysis of pre-treatment tumours in our cohort suggested a greater abundance of *Fusobacteria* in intermediate and poor responders. In addition, we identified cases in which microsatellite instability developed or was selected for during exposure to NACRT.

## Figures and Tables

**Figure 1 cancers-12-01808-f001:**
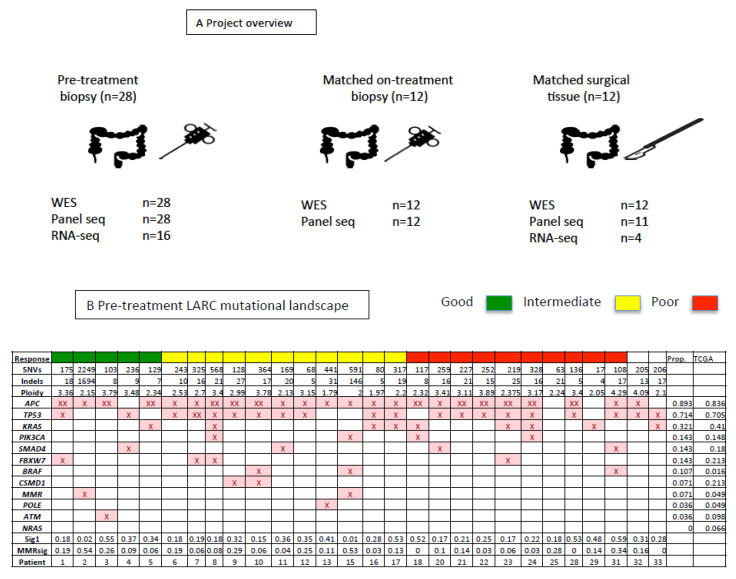
Project overview and pre-treatment genomic landscape. (**A**) Project overview—number of patients from whom pre-treatment, matched on-treatment, and matched resection tumour samples were collected. (**B**) Summary of genomic mutations identified in pre-treatment patient tumours by response category. Number of single nucleotide variants (SNVs), indels, overall ploidy and mutations in known colorectal cancer genes are shown. The proportion of mutations attributed to COSMIC mutational signature 1 (Sig1) and mismatch repair-associated signatures (MMRsig) are shown. (**C**) Frequency of copy number alterations (CNAs) by chromosome arm in the pre-treatment tumours. (**D**) Log 2 ratios of colorectal cancer genes known to be affected by CNA. Red indicates positive log 2 ratio (gains) and blue indicates negative log 2 ratio (losses).

**Figure 2 cancers-12-01808-f002:**
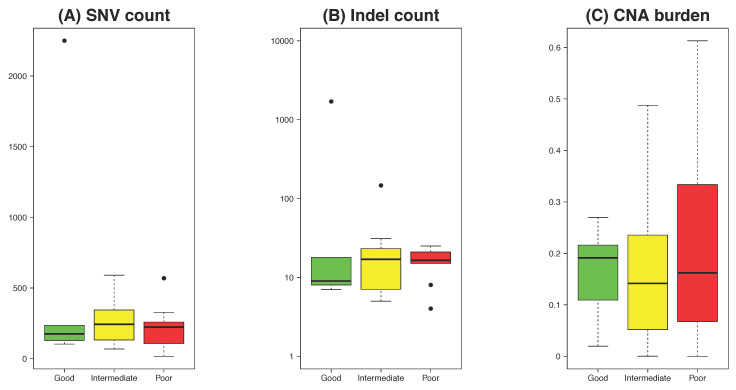
Genomic and transcriptomic landscape by treatment response category. (**A**) Single nucleotide variants (SNV) count. (**B**) Indel count. (**C**) Copy number alteration burden as a proportion of the genome. (**D**) Frequency of copy number alterations (CNAs) by chromosome arm. (**E**) Microbial abundance identified in RNA-seq data. (**F**) Immune composition of B cells, macrophages, neutrophils, NK cells, CD4^+^ T cells, CD8^+^ T cells, and regulatory T cells identified from RNA-seq data.

**Figure 3 cancers-12-01808-f003:**
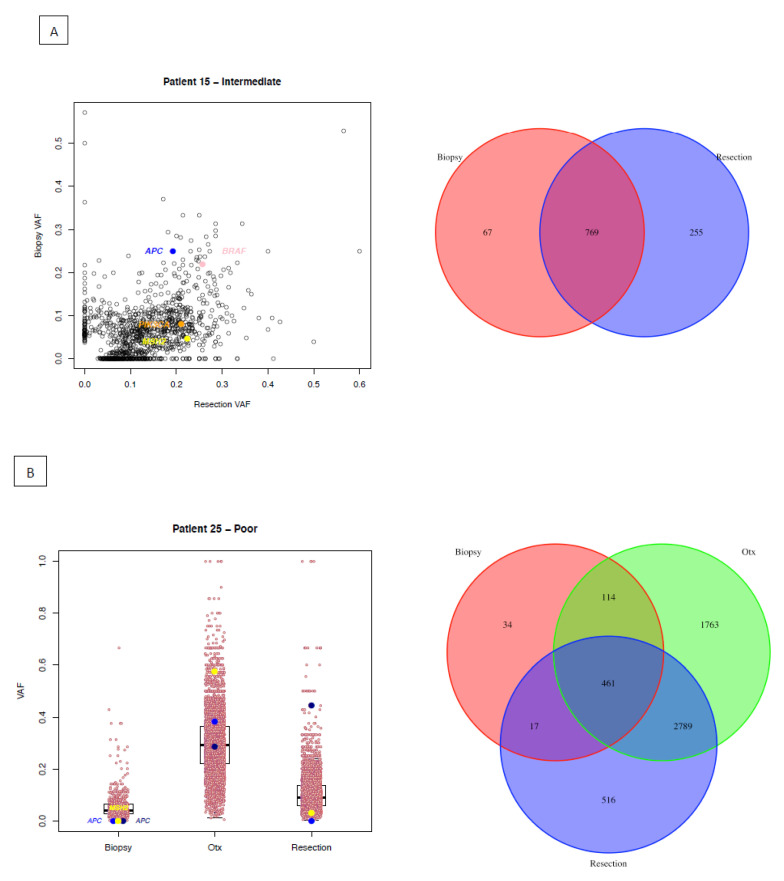
Patients with potential selection of microsatellite instability. (**A**) Patient 15. Left panel: Scatterplots comparing Variant Allele Frequencies (VAFs) of all SNVs (black circles) and likely driver mutations (coloured circles) in the pre-treatment biopsy and post-treatment surgical resection tumour sample. Right column: Venn diagram of all SNVs from different time points. (**B**) Patient 25. Left panel: Boxplots of Variant Allele Frequencies (VAFs) of all SNVs (black circles) and likely driver mutations (coloured circles) in multiple tumour samples. Right column: Venn diagram of all SNVs from different time points.

## Data Availability

The data that support the findings of this study are available on request from the corresponding author. The data are not publicly available due to privacy or ethical restrictions. Data will be available on request upon publication from the European Genome-phenome Archive (EGA) database by contacting the data access committee (Genomic Oncology Research Group DAC: EGAC00001001585) assigned for this project.

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
