# Peer review of "Genomic and Transcriptomic Characterisation of Response to Neoadjuvant Chemoradiotherapy in Locally Advanced Rectal Cancer"

_cancers, 2020, doi:10.3390/cancers12071808_

Round 1
Reviewer 1 Report
Toomey et al described the transcriptome and genomic profile of 33 locally advanced rectal carcinomas (LARC). Predictive markers of response to neoadjuvant therapy (NACRT) are necessary, considering that the current management resulted in a partial response in 70-90% of patients and about 20% show resistance to treatment. The investigation of pre, during, and post-treatment has a relevant impact in the area. The manuscript is potentially interesting but several issues need to be appropriately addressed.
- The authors used samples (N=33 as presented in the Abstract section and Patient Characteristics - lines 75-87) collected before, during, and after NACRT to perform whole-exome sequencing, transcriptome, and target sequencing (two distinct panels with few genes overlapping). It is confusing to follow how many cases were evaluated using matched samples. It is stated that they analyzed 33 samples (lines 32-33); however, there are incongruencies in this number (see lines 124-125 shows 26 instead of 28). Figure 1A should be better designed showing the strategies used in matched samples.
- Overall the findings were poorly explored and presented. Although the authors showed several figures, the tables with the detailed results are missing. For instance, they reported a comparison with the TCGA data. It is not clear if they used LARC samples or the entire cohort of rectal cancer cases. What were the criteria used to select the TCGA cases?
- Sequencing data results using gene panels included in this study: Why did they use different gene panels (with few overlapping: 48 and 151 genes)? How many samples were evaluated, and what are the selection criteria to perform this strategy? How were these cases selected?
- Copy number alterations: Figures and 1C and 2D showing the summary of the genomic alterations in graphs are not sufficient to detect differences among the groups of tested cases. The results are superficially presented. They also discussed the gains of genes mapped in chromosome 20 (lines 258-263). However, these results were not clearly demonstrated. The statistical analysis of the comparison among the groups tested is missing.
- Transcriptomic data: Apparently, they used the findings exclusively to explore the microbiota and the immune signature. Did the authors compare their results with data already published in LARC evaluated according to response to therapy? The published papers have described array platforms instead of RNAseq. However, the gene lists are available, and the authors could easily perform this analysis.
- Mutational analyses: a recent study described mutations in genes related to homologous recombination in LARC patients evaluated according to response to therapy (Front Oncol. 2019 May 14;9:395. doi: 10.3389/fonc.2019.00395. eCollection 2019.PMID: 31192117). This analysis could be helpful for this study as well as a comparison with others in LARC. Also, a table having these results should be provided.
- Microbiota analysis: these results are very summarized.
- Discussion section: lines 220-225 - What is your explanation for these findings?
- The limitations of the study should be better presented.
Reviewer 2 Report
The authors report a study on molecular characterisation of tumour response to neoadjuvant chemoradiotherapy (NACRT) in locally advanced rectal cancer.
This issue is of a great clinical relevance because of the prognostic impact of the pathological complete response and its implications on conservative surgical strategies including organ preservation. The definition of molecular factors in the prediction of pathological complete response is therefore an important clinical need.
The pre-treatment genome was not predictive of tumour response to NACRT. However, RNA analysis of pre-treatment tumours suggested a greater abundance of Fusobacteria in intermediate and poor responders. In addition, a preliminary evidence of selection for mismatch repair deficiency in response to NACRT was reported.
The authors conclude that predictors of response to NACRT are not likely to be determined simply by genomic analysis and overall response could be determined by a multitude of factors that are yet to be elucidated
This is a very interesting study, although done in a small cohort of patients. I have some additional comments regarding the report:
-The study was conducted on two patient cohorts deriving from two different centers. The DNA targeted panel is different in term of genes number (151 vs 48) and median coverage (368x vs 5000x). The sensitivity in the mutation detection is therefore very different (more than 10 times), and also the panel of genes monitored. How are these differences taken into account? How many samples were derived from each center?
-How was the tumor samples dissected to obtain the tumor fraction? Was a minimum threshold of tumor cellularity defined?
-When looking at supplementary Figure 1 it looks like there is a trend for a decrease of the allelic fraction for all the somatic mutations. Could this be related to a decrease of the tumor cellularity or is it normalized?
-The authors stated that no difference was observed in the mutational tumor burden between pre- and on-NACRT (lines 154-155). It should be clarified how this parameter has been calculated based on the analysis of small targeted panels and how they can compare results from the two cohorts investigated with different panels and at different coverages.
-Considering the simultaneous availability of SCNAs and transcriptomic data on the same samples I would be curious to see if any relationship exists between those data to highlight the effect of chromosomal gains and losses on the transcriptional phenotypes.
Other minor comments include.
- Time to surgery after NACT should be reported, because of its impact on tumour response
